# PiCCO or Cardiac Ultrasound? Which Is Better for Hemodynamic Monitoring in ICU?

**DOI:** 10.3390/medicina60111884

**Published:** 2024-11-17

**Authors:** Maria Andrei, Nicoleta Alice Dragoescu, Andreea Stanculescu, Luminita Chiutu, Octavian Dragoescu, Octavian Istratoaie

**Affiliations:** 1Department of Anesthesiology and Intensive Care, Emergency County Hospital of Craiova, Faculty of Medicine, University of Medicine and Pharmacy of Craiova, 200349 Craiova, Romania; 2Department of Urology, Emergency County Hospital of Craiova, Faculty of Medicine, University of Medicine and Pharmacy of Craiova, 200349 Craiova, Romania; 3Department of Cardiology, Emergency County Hospital of Craiova, Faculty of Medicine, University of Medicine and Pharmacy of Craiova, 200349 Craiova, Romania

**Keywords:** hemodynamic monitoring, echocardiography, septic shock, cardiogenic shock

## Abstract

Advanced hemodynamic monitoring is fundamental in the management of the critically ill. Blood pressure and cardiac function are key markers of cardiovascular system function;, thus, having accurate measurements of these parameters in critically ill patients is essential. Currently, there are various methods available to choose from, as well as a greater understanding of the methods and criteria to be able to compare devices and select the best option for our patients’ needs. Cardiac ultrasound and transpulmonary thermodilution help tailor the therapy for a patient’s individual needs by putting the results of a thorough hemodynamic assessment into context. Both these hemodynamic monitoring techniques have their advantages, drawbacks and limitations. Cardiac ultrasound is a safe, non-invasive, less expensive, efficient bedside tool for diagnosing, monitoring and guiding critically ill patients’ therapy management. It is recommended in the consensus guidelines as the first-choice method, especially when it comes to identifying different types of shock or the various factors involved. Pulse index contour continuous cardiac output (PiCCO) is a minimally invasive hemodynamic monitoring technique, integrating various static and hemodynamic parameters through a combination of trans-cardiopulmonary thermodilution and pulse contour analysis. The PiCCO method provides guidance to fluid and vasoactive therapy in critically ill patients and is also used for intraoperative and postoperative fluid management and monitoring in cardiac surgery. While invasive methods such as PiCCO are recommended for hemodynamic monitoring and can provide accurate information, they are not always necessary and are contraindicated in some cases.

## 1. Introduction

Advanced hemodynamic monitoring is fundamental in the management of the critically ill with circulatory failure. Nowadays we have at the ready a number of alternatives for monitoring cardiac output as well as a greater understanding of the methods and criteria to be able to compare devices and choose the best option for our patients’ needs.

Shock is a life-threatening condition defined as inadequate organ and peripheral tissue perfusion [1]. Cardiac output is the main factor of global oxygen delivery to organs and a sensitive marker of cardiovascular failure. Both vascular and cardiac functions may be rapidly altered; thus, repeated measurements of cardiac output should be made in order to be able to characterize the type of shock, select the appropriate therapy and evaluate the patient response [2].

Septic shock is the primary cause of cardiovascular failure in the ICU [3]. The Third International Consensus Committee defines sepsis as “life-threatening organ dysfunction caused by a dysregulated host response to infection” [3]. Septic shock is defined by the circulatory, cellular and metabolic abnormalities in the setting of sepsis. In septic shock, there are several hemodynamic abnormalities that occur. The cardiovascular system is altered at macrocirculatory (hypovolemia, vascular tone depression, myocardial dysfunction) and microcirculatory (altered microcirculation, impaired oxygen extraction) levels [4]. Septic patients present fluid-refractory hypotension requiring vasopressor therapy to maintain a mean arterial pressure (MAP) of ≥65 mmHg and prevent associated tissue hypoperfusion, typically evidenced by a serum lactate > 2 mmol/L [5].

Another cause of cardiovascular failure in the ICU is cardiogenic shock. Cardiogenic shock is defined by the existence of a primary cardiac disease leading to hypotension (systolic blood pressure < 90 mmHg, or vasoactive drugs needed to maintain a systolic blood pressure ≥ 90 mmHg) and signs of impaired perfusion (altered mental status, oliguria, cold extremities and clammy skin, arterial lactate > 2 mmol/L) in the state of normovolemia or hypervolemia [6,7]. Acute myocardial infarction (AMI), or ST-elevation myocardial infarction (STEMI), is the primary cause of cardiogenic shock, but there are other causes, including acute decompensated heart failure, massive pulmonary embolism, myocarditis, severe valvular dysfunction, etc. By etiology, cardiogenic shock can be divided into two major branches: acute coronary syndrome-related cardiogenic shock and non-acute coronary syndrome-related cardiogenic shock [6,7].

Sequential hemodynamic monitoring is paramount for the accurate diagnosis, staging, risk stratification and management of both septic and cardiogenic shock. There are various forms of monitoring, both invasive and non-invasive, that identify patients requiring therapy and/or mechanical support, as well as assess responsiveness and thereby assist with mortality reduction [7].

In this review, we detail different monitoring methods, including echocardiography and transpulmonary thermodilution, the relevant parameters to observe and the way they can be utilized to sustain the effective management of critically ill patients and also improve patients’ outcomes.

Both these hemodynamic monitoring techniques have their advantages, drawbacks and limitations. One size does not fit all; thus, echocardiography and transpulmonary thermodilution should help tailor the therapy for a patient’s individual needs by putting the results of a thorough hemodynamic assessment into context [2,8,9].

## 2. Septic Shock

Septic shock has a high mortality risk associated with organ dysfunction, and therefore early recognition, efficient and timely source control and fluid resuscitation are vital to patient management [5]. Septic shock resuscitation includes early aggressive fluid administration and use of vasoactive drugs to improve systemic blood flow and restore tissue perfusion. However, an important consideration is avoidance of over-resuscitation, which can lead to an increase in morbidity and/or mortality. This requires delicate management, with the goal to stop resuscitation when normal physiological values indicate reestablished tissue perfusion. It is important to mention that there is no absolute value that can be considered normal and pursued as the target reference during resuscitation and that macrocirculatory values should be amended to meet perfusion requirements [5,10,11,12].

Pathophysiologically, septic shock can be divided into two phases [5]. The early stage is related to loss of vascular tone, which leads to hypotension by arterial vasodilatation and also to a decrease in venous tone. Therefore, stressed volume, mean systemic filling pressure and the driving pressure for venous return are declining, resulting in a drop in preload (hypovolemia) and cardiac output. All these lead to a decline in systemic blood flow and mean arterial pressure (perfusion pressure). As a compensatory mechanism, vasoconstriction improves venous return and elevates mean arterial pressure (MAP) on the arterial side. Early fluid resuscitation is used to address hypotension and increase mean arterial pressure. Use of vasoactive drugs such as epinephrine and norepinephrine also increases cardiac output; however, prolonged therapy is associated with negative effects such as myocardial over-stimulation and hepato-splanchnic vasoconstriction. Thus, a physiological state is attained through a balance between micro- and macrocirculation, known as hemodynamic coherence [5,13].

In the late stage of septic shock, hemodynamic coherence is lost with associated metabolic, endothelial and coagulation dysfunction [5]. These alterations further disturb microcirculatory flow and impact tissue and cellular oxygenation, leading to hypoxia. Fluid overload further impairs the modifications, leading to an upsurge in venous congestion, interstitial edema and diffusion distance for oxygen [4,5,11,13].

Given these pathophysiological changes in septic shock, it is essential to accurately determine and monitor the underlying cardiovascular changes. Clinical examination by itself is not enough to elucidate the predominant component that leads to circulatory dysfunction (hypovolemia, vascular tone depression, myocardial depression, altered microcirculation, impaired oxygen extraction), which is why other markers need to be attained through non-invasive or invasive techniques. Having at disposal a variety of hemodynamic markers helps to weigh the efficacy of hemodynamic procedures while also concluding with the designation of the most beneficial therapy [2,4,13].

The usage of an arterial catheter in patients with shock is highly advisable for monitoring in real time the arterial pressure, as well as for receiving pertinent information about other markers such as systolic arterial pressure, diastolic arterial pressure, mean arterial pressure and pulse pressure, all pertaining to cardiovascular status [4].

Systolic arterial pressure (SAP) is one of the most trusted variables for best determining the left ventricular afterload [4]. But while the SAP at central vascular level is low, in the periphery it is higher due to pulse wave amplification (this phenomenon is depreciated in elderly patients), thus, in the femoral artery there is a lower SAP determined than in the radial artery [4].

Diastolic arterial pressure (DAP) estimates the upstream pressure for left ventricle perfusion while also being a marker of arterial tone. Low arterial tone is often linked with a lower DAP, usually evoked by bradycardia or serious arterial stiffness. A low DAP in a patient diagnosed with shock, who has a normal or higher heart rate, is a marker for decreased arterial tone and is thus paramount for determining the form and state of shock and also the right time to start early vasopressor therapy. A decreased DAP can lead to myocardial ischemia in patients with coronary artery disease, as well as requiring vasoactive treatment. DAP determined in the femoral artery is higher than in the radial artery [4,13].

Mean arterial pressure (MAP) displays the upstream pressure for the vital organs’ perfusion and usually is consistent in the arterial bed and slightly higher in the aorta correlated to the periphery. Physiologically, MAP is higher than the central venous pressure, which displays the downstream pressure for organ perfusion, consequently designating MAP as the organ perfusion pressure. Nevertheless, in cardiogenic shock, MAP is low and CVP is high; in this case, MAP underrates the organ perfusion pressure, thus making the difference between these two parameters (MAP-CVP) the organ perfusion pressure [4,12,13,14].

Pulse pressure (PP) at the aortic level is based on stroke volume and arterial stiffness. In elderly patients, PP is usually increased, discounting the case when the stroke volume is low. Therefore, a normal PP (40–50 mmHg) and a lower PP are markers for a low stroke volume [4].

Another marker relevant for septic shock patients is mixed venous blood oxygen saturation (SvO2), which is complementary to arterial blood oxygen saturation (SaO2), oxygen consumption (VO2), cardiac output and hemoglobin concentration, designated using the Fick equation applied to oxygen (SvO2 = SaO2 − [VO2/ (cardiac output × Hb × 13.4)]). Therefore, SvO2 is appointed as a marker that indicates the balance between DO2 and VO2, with normal values at 70–75%. Because monitoring SvO2 necessitates a pulmonary artery catheter, a device that is becoming out of date, another marker has been nominated to replace it: central venous blood oxygenation saturation (ScvO2). However, ScvO2 is no longer recommended as a therapeutic target by the Surviving Sepsis Campaign [4]. Still, this marker may have its use for adjusting resuscitation therapy management. Normal or higher values of ScvO2 denote that oxygen extraction capacities are impaired, so an increase in DO2 will not correct tissue hypoxia. Lower values of ScvO2 denote that DO2 is insufficient, and therapeutical assistance is needed to raise it by increasing cardiac output [4,5,12,13,14].

Cardiac ultrasound is extremely useful as a tool for hemodynamic monitoring by quickly appraising the systolic and diastolic function of both ventricles. In the management of critically ill patients, the most used ultrasound markers are the left ventricular ejection fraction (LVEF), the velocity time integral (VTI), the left ventricular size, the right ventricular end-diastolic area (RVEDA)/left ventricular end-diastolic area (LVEDA), the early wave of transmitral diastolic blood flow (E), the atrial wave of transmitral diastolic blood flow (A), the tricuspid annulus systolic excursion (TAPSE), the pulmonary artery systolic pressure (PAPs) and the inferior vena cava diameter [1,2,4,15]. Furthermore, other anomalies that lead to circulatory failure, like pericarditis and cardiac valve diseases, can be quickly discovered. Also, echocardiography can estimate preload responsiveness [4].

The ESAIC has recommended the transpulmonary thermodilution technique for monitoring patients with circulatory shock associated with ARDS [4]. These hemodynamic devices provide intermittent values for cardiac output. However, these results highly depend upon metabolic conditions. The most important markers to monitor are extravascular lung water (EVLW) and the pulmonary vascular permeability index (PVPI), used for the appraisal of pulmonary edema and the state of lung capillary leakage. Other markers to pay attention to are the global end-diastolic volume (reflects global cardiac preload) and the cardiac function index (derived estimate of systolic heart function) [4,7].

## 3. Cardiogenic Shock

Cardiogenic shock is defined by the existence of a primary cardiac disease leading to hypotension (systolic blood pressure < 90 mmHg, or vasoactive drugs required to maintain a systolic blood pressure ≥ 90 mmHg) and signs of impaired perfusion (altered mental status, oliguria, cold extremities and clammy skin, arterial lactate > 2 mmol/L) in the state of normovolemia or hypervolemia [6,7]. Hemodynamic parameters such as cardiac index (CI) < 2.2 L/min/m² and pulmonary capillary wedge pressure (PCWP) > 15 mmHg might help confirm the diagnosis [7]. According to pathophysiology, cardiogenic shock can be divided by degree: pre-shock (despite increased peripheral vascular resistance and systolic blood pressure > 90 mmHg, there are signs of hypoperfusion); shock phase (there are both hypoperfusion and hypotension present); and refractory shock (hypotension is unresponsive to therapy), further detailed in Table 1 [6]. By etiology, cardiogenic shock can be divided into two major branches: acute coronary syndrome-related cardiogenic shock and non-acute coronary syndrome-related cardiogenic shock. Establishing the patient’s stage is essential to providing the best management [6,7,14,16].

The management of cardiogenic shock entails diagnosing the concealed pathology that started the chain of events. If either hypoxemia or respiratory distress is present, early rapid provision of respiratory support, including mechanical ventilation and hemodynamic support therapy with inotropic and/or vasopressor agents, is required [14].

Humoral markers that might predict an adverse outcome in cardiogenic shock are raised plasmatic arterial lactate as an early marker for mitochondrial dysfunction and cellular damage and creatinine levels and increased transaminase as markers for renal and hepatic hypoperfusion. Also, acidosis is another marker that conveys negative effects on myocardial contractility, which may hinder the response to some vasopressor drugs [6,14].

Ultrasound is by far the quickest at-the-ready first-line tool for a thorough evaluation of critically ill patients, helping to determine a potential cause for cardiogenic shock from the various disorders that could lead to it. As a non-invasive monitoring tool, cardiac ultrasound can help with differential diagnosis, especially since the shock could be multifactorial, such as a combination of cardiogenic and septic shock. Also, clinicians have at their disposal rapid hemodynamic and fluid status assessment, thus prompting an effective management strategy and further providing a reliable tool to monitor progress [17].

Another useful tool at clinicians’ disposal is minimally invasive hemodynamic monitoring, such as the PiCCO system. While basic vital signs like heart rate, blood pressure and oxygen saturation can be monitored using non-invasive devices, for a critically ill patient, other parameters like cardiac output and systemic vascular resistance are essential, and, with the help of PiCCO, continuous hemodynamic monitoring is provided, helping to guide the therapy [8].

Table 2 contains a summarized comparison of both the PiCCO method and the cardiac ultrasound technique that will be further detailed.

## 4. Hemodynamic Monitoring of Critically Ill ICU Patients

Although the gold standard in practice in most ICU settings is the use of a pulmonary artery catheter (PAC), complications like bleeding, infection and dependence on good peripheral vasculature, as well as its expense and limited availability, restrict its use. Also, due to its highly invasive nature, the PAC is no longer the most used device for continuous hemodynamic monitoring [2]. While the preference is for gold standard monitoring achieved in a non-invasive manner (see below), newer systems such as PiCCO rely on limited invasive arterial lines to provide comparable information regarding hemodynamic measurements [2,8].

Characteristics of the ideal hemodynamic monitoring device [2]:-Determines significant parameters to therapy management.-The value of the measured parameter is comparable to the standard reference.-Repeated measurements determine similar results under stable conditions.-Updates parameters to display the current clinical status.-Non-invasive tool, with no risk to the patient when installed.-Quick and easy to install and use with few components and minimal training necessary.-Inexpensive acquisition and maintenance.-Similar to other monitoring devices and parameters.-Useful to different specialties and various clinical conditions.-Demonstrated contribution in clinical trials.

## 5. Cardiac Ultrasound

Cardiac ultrasound is a non-invasive, quick-to-perform, cheap and efficient tool for diagnosing, monitoring and guiding critically ill patients’ therapy management, as depicted in Figure 1. It is recommended in the consensus guidelines as the first-choice method. The clinicians need to perform a careful, thorough examination, even in an emergency situation where a rapid appraisal is required [18].

Cardiac ultrasound covers a broad range of diagnostics that no other bedside device can, from underlying cardiac and hemodynamic issues in the left and right heart to fluid alterations and cardiac response to vasoplegia found in septic shock. Echocardiography is the only technique that can help diagnose the latent cause for many pathologies leading to severe cardiac failure, including sepsis [1,2,14,17].

Cardiac abnormalities in severe sepsis include the following:-Left ventricular dilatation.-Left ventricular contraction alteration.-Left ventricular diastolic dysfunction.-Right ventricular systolic/diastolic dysfunction.-Ventricular outflow obstruction.-Valvular lesions (functional, endocarditis).

In terms of technique availability, using a basic-level 2D transthoracic echo and M-mode in five views elucidates serious pathologies in the acute phase (LV/RV contraction, ejection fraction, segmental wall motion abnormalities, intravascular fluid status, pericardial tamponade). However, the more advanced Doppler techniques, like spectral Doppler and tissue Doppler (TDI), provide both diagnostic and hemodynamic evaluation, proving to be more inclusive (LV/RV systolic function, diastolic function, valve structure/function, intravascular fluid status, pericardial tamponade, pulmonary artery pressure, left atrial pressure, cardiac output, ventricular outflow obstruction) [1,2,4,16].

The left ventricular output tract (LVOT) VTI as a single parameter can be used to indicate the stroke volume, with a normal value of >20 cm. A measurement of over 18 cm estimates an appropriate stroke volume. Other parameters for overall cardiac function are the myocardial performance index and mitral annulus plane systolic excursion [1,4].

The most common cause of cardiogenic shock derives from a marked decrease in left ventricular contraction [1]. Contractility is the aptitude of the myocardium to contract against a specific load or any given preload. What the echography measures is the degree of the myocardial fiber shortening that happens during systole. Measurements of both the left atrium and ventricle may elucidate the duration of the contractile dysfunction, while observing dilation signals a chronic phase of the disease. Any examination of left ventricular contractility needs to ponder whether segmental wall motion abnormalities are present, and, if so, urgent revascularization is an option to ponder over [1]. Left ventricular ejection fraction (LVEF) is a parameter that can serve as a prognostic marker for chronic heart failure patients diagnosed with cardiogenic shock [1]. Interpretation needs to consider the effects of arterial blood pressure (afterload, use of vasoactive drugs), because an impaired left ventricle may appear normal in the presence of inotropes. Also, other cardiac pathologies need to be considered because marked diastolic or valvular dysfunction can be hidden while the LVEF appears to be normal or high [1,2,16].

Using the cardiac ultrasound technique also helps with discovering possible valvular abnormalities, both acute or chronic, such as mitral regurgitation or degenerative aortic stenosis [1,2,7,19], commonly diagnosed in elderly patients [1].

The assessment of left ventricular function is another task echocardiography is useful for. Critically ill patients with hypermetabolism have compromised diastolic function. Using the TDI technique, the clinician can appraise the mitral annulus for an adequate assessment of left atrial pressure (LAP), a sound marker for left ventricular function and preload [1]. It is important to note that elevated left atrial pressures differ between patients on positive ventilation and non-ventilated patients, with lower values in the former. Positive pressure ventilation affects left ventricular diastolic filling, often in an opposite way. An elevated intrathoracic pressure decreases systemic venous return, leading to decreased left ventricular preload and also a lower atrial-ventricular pressure gradient. Lung hyperinflation can reduce pulmonary vascular resistance when the volume increase is lower than the functional reserve capacity; otherwise, it will increase the resistance, affecting both the right ventricular afterload and the left ventricular preload [1,17].

Inferior vena cava (IVC) diameter measured on a cardiac ultrasound in the subcostal view, ~0.5–3 cm from the cava–right atrial junction, is used for the appraisal of fluid responsiveness (reference). In the acute phase of shock, an IVC diameter for a spontaneously breathing patient of <10 mm indicates fluid therapy responsiveness, while a diameter of >20 mm suggests that the patient is unlikely to respond. In addition to its use for determining fluid responsiveness, IVC variation can also provide an estimate of the right atrial pressure (RAP). An IVC diameter of <21 mm measured in a spontaneously breathing patient that collapses > 50% with a sniff indicates a normal RAP of 3 mmHg, while an IVC diameter of >21 mm that collapses > 50% with a sniff points to an RAP > 15 mmHg [1].

Left ventricular wall hypertrophy signals to the clinician to consider hypertrophic obstructive cardiomyopathy; thus, searching for left ventricular outflow obstruction is considered standard practice, although a dynamic LVOT obstruction can be present in elderly people without wall hypertrophy being present. Critically ill patients are prone to being diagnosed, and, aside from age, other accountable factors are tachycardia, hypovolemia and inotrope drugs [1].

Cardiac ultrasound is a remarkable tool, extremely useful in the diagnosis and management of shock, especially where etiology is similar or multifactorial. It is a non-invasive technique, is rapid to initiate and is an easily accessible bedside tool that can be utilized any time of the day or night. On the other hand, there are also drawbacks and limitations to using this technique. It is operator dependent, and it requires a significant amount of training and experience for a clinician to develop echography aptitudes to provide a comprehensive examination and accurately diagnose a pathology, on which therapy management relies. There are also high start-up costs. And even though there is an easily accessible device, applying the technique several times a day is time-consuming. There is also the matter of continuous monitoring, which, ideally, aside from utilizing a non-invasive tool, we also desire, especially for monitoring critically ill patients [1,2,9,14,16,17].

## 6. Pulse Index Continuous Cardiac Output Measurements (PiCCO)

PiCCO is a technique utilized to monitor cardiac output. Utilizing the PiCCO monitor is a minimally invasive hemodynamic monitoring technique, integrating various static and hemodynamic parameters through a combination of trans-cardiopulmonary thermodilution and pulse contour analysis. Pulse contour analysis estimates that the contour of the arterial waveform is proportional to the cardiac stroke volume. It consists of three main elements: an arterial catheter with a solid-state thermistor from its tip, an injection device that attaches to a central venous catheter and the user-interface touchscreen monitor, which displays the clinical parameters, as depicted in Figure 2 [8,9,20].

One of the cases in which the PiCCO method is utilized is for monitoring critically ill patients and for providing guidance to fluid and vasoactive therapies; other purposes are for intraoperative and postoperative fluid management and monitoring patients during cardiac surgeries [9,20,21,22].

PiCCO may display imprecise readings in one of these cases: intracardiac shunts, aortic aneurysm, pulmonary embolus, partial lung resection, unstable arrhythmias. There are also contraindications to using this method, such as severe peripheral vascular disease, arterial grafting, the presence of local infection and coagulopathy and anatomical or physiological alterations that can significantly alter the measurements [9,14].

Different figures for arterial pressure may be determined when the arterial catheter is placed in the peripheral compared to the central site; also, in this case the shape of the arterial wave pressure may vary. The most common site for placing the arterial catheter is the femoral artery [9] but there are other options available, such as the radial, brachial and axillary arteries. The procedure for insertion requires a full aseptic technique, as for central venous cannulation. The venous catheter should be situated in the central cardiopulmonary circulation in the proximity of the right atrium. Electing the femoral vein site will lead to overestimating the intrathoracic volumetric figures, even though the cardiac output data may be similar [9,20]. The manufacturer recommends using the arterial catheter for up to 10 days, but if complications appear, it needs to be removed immediately. The monitoring kits (the arterial line transduction kit and inline injectate sensor housing) are to be replaced every four days. Complications can also appear post-cannulation, such as bleeding, hematoma, signs of infection, perfusion impairment and the misplacement of the catheter, in which case the catheter needs to be removed. After all these steps are accomplished, calibration needs to be performed before data acquisition. The injectate solution should ideally be 0.9% sodium chloride at an optimum temperature of <11 °C, while the injected volume derives from the patient’s weight and the extravascular lung water (EVLW). Usually, three separate injections of 15–30 mL saline are administered. Calibration should be repeated at 8 h intervals or whenever a major change in the patient’s clinical condition appears [9,14,20,23].

Trans-cardiopulmonary thermodilution provides the intermittent estimation of parameters such as cardiac output (CO), ELVW, global end-diastolic volume (GEDV), intrathoracic blood volume (ITBV), cardiac function index (CFI) and global ejection fraction (GEF). Pulse contour analysis continuously measures cardiac output, stroke volume, stroke volume variation, pulse pressure variation and an index of left ventricular contractility [9,14,20,24]. Other parameters measured using the PiCCO method are basic parameters such as heart rate, systolic, diastolic and mean arterial blood pressure; central venous pressure; and central venous oxygen saturation, and these have been already described in the beginning of this review [4,9,12,13,14].

CO is a key determinant of tissue perfusion. This parameter derives from the Stewart–Hamilton equation. There is a great advantage in the ability to measure cardiac output through thermodilution, dispensing with the need for right atrium catheterization. Under normal conditions, CO varies in order to comply with the total tissue requirements, which are susceptible to change in case of infection, heart disease, administration of drugs, etc., and it depends on the metabolism; therefore, CO can be low but sufficient when low oxygen is required, or it can be high but insufficient when oxygen requirements have risen. When the cold infusion passes through the heart and the lungs, aside from CO, other parameters such as preload and extravascular lung water are measured. CO and stroke volume can also be determined through pulse contour analysis, but the measurement performance of this method can be impaired when vascular tone is altered [9,22].

EVLW and the pulmonary vascularity permeability index (PVPI) represent quantitative parameters for the degree of lung capillary leakage and the volume of pulmonary edema. Potential sources of error include lung resection, obstruction of major pulmonary vessels and possibly a high positive end-expiratory pressure. EVLW is used as a marker to guide personalized fluid management, but it may also have a role in predicting the survival of critically ill patients. Furthermore, EVLW may elucidate an underlying sub-clinical pulmonary edema, concealed on clinical examination or plain radiography, and also differentiate between the hydrostatic and inflammatory causes of pulmonary edema. It is recommended to calculate the EVLW by using an idealized body weight, since using the actual body weight leads to an underestimation in obese patients or an overestimation in underweight patients [9].

GEDV and ITBV are volume-based markers of preload and volume responsiveness measured using PiCCO. GEDV is a volume that derives from the hypothesis that all four heart chambers are simultaneously in the diastolic phase. GEDV represents the difference between the intrathoracic thermal volume (ITTV) and pulmonary thermal volume (PTV). A GEDV index (GEDVI) measured with the PiCCO method better epitomizes the echocardiographic changes in left ventricular preload in response to fluid resuscitation than the continuous end-diastolic volume index (CEDVI) measured by a modified PAC. The CFI and GEF are derived as ratios of cardiac output and volume-based measurements. However, their use in clinical practice is not reliable, since derived parameters are often altered by measurement errors, and the individual components are already available [9,24].

In mechanically ventilated patients, heart–lung interactions can be used to accurately determine the response to fluid therapy using dynamic markers compared with pressure- and volume-based parameters. These parameters, stroke volume variation (SVV), pulse pressure variation (PPV) and systolic pressure variation (SPV), can be determined using pulse contour analysis. The basic concept is that if the heart is preload responsive, the modifications in preload determine the stroke volume to adjust with the ventilation [4,9,21,22,23].

Pulse pressure is directly proportional to the left ventricular stroke volume. Pulse pressure variation reflects the changes in the peripheral pulse pressure during the respiratory cycle and for that reason also represents a parameter for measuring stroke volume variation. Although a variation of between >10 and 13% predicts a higher cardiac output in response to fluid challenge, there are various factors that interfere with PPV assessment of fluid responsiveness, such as the presence of cardiac arrhythmia, low tidal volume ventilation, spontaneous ventilatory effort and low respiratory compliance [4,9,22,23]. To bypass these inconveniences, researchers have recommended gradually increasing the tidal volume from 6 mL/kg to 8 mL/kg to appraise the dynamic response of PPV. The results are remarkably good for critically ill patients in supine and in prone positions and also in patients undergoing surgery. In critically ill patients, the use of PPV and other fluid responsiveness markers is recommended by the sepsis guidelines when possible, thus making PPV a useful marker for guiding fluid resuscitation [4,22,23].

The greatest advantage of the PiCCO method is its proficiency in measuring and integrating into clinical decision-making a multitude of hemodynamic validated parameters, which is beneficial for critically ill patients’ therapy management. It is minimally invasive, since most critically ill patients usually already require the placement of a central venous catheter and arterial line. The placement of a central venous catheter and arterial line provides continuous measurement. GEDV and ITBV give a better estimation of preload than CVP and PCWP and are not influenced by mechanical ventilation. EVLW has demonstrated a clear correlation in the severity of ARDS, number of ventilator days, ICU duration and mortality [20,24].

There are also drawbacks to using this technique, including the following: recalibration is required with changes in position, therapy or condition to account for compliance of the vascular bed; in obese patients, EVLW is underestimated as related to the patient’s weight, and IBW needs to be considered; EVLW is only measured in parts of the lung that are perfused and is underestimated post pneumonectomy; and severe arrhythmias may lead to an inaccurate thermodilution washout curve [9]. Aside from cardiac and extracardiac abnormalities that interfere with accurate measurements, there are also the following errors of technique: the temperature and the volume of the injectate should be consistent between measurements; catheter tip malposition, which should be placed in an area with good blood flow; misplaced thermistor either against the vessel wall or near to the source of injectate; the measurement should be performed at end-expiration, and measurements performed at random times in the respiratory cycle should be avoided. There are also risks associated with cannulation and ongoing use, such as hematoma, ischemia and infections [8,9,14,19,20,25].

## 7. Conclusions

There are various methods of advanced hemodynamic monitoring available today. Selecting one of them requires an understanding of its physiological basis, the risks and complications associated with the use of the respective device, as well as the range and validity of the data provided. These factors must be regarded within the clinical situations of recommended use and should weigh the efficacy of estimated parameters, which will be employed in the clinician’s decision-making process, as well as the potential implications of those decisions.

Cardiac ultrasound is perhaps the most useful and safe method in the diagnosis and management of shock, especially when it comes to identifying different types of shock or the various factors involved. This method has fewer complications, is less expensive and can be used for the thorough hemodynamic monitoring of a critically ill patient’s status. Invasive methods are still recommended for monitoring critically ill patients, providing accurate information; nevertheless, they are not always necessary, being relatively contraindicated in some cases. Non-invasive methods are reliable, and their use is increasing, especially with advancements in the newer generation of devices.

## Figures and Tables

**Figure 1 medicina-60-01884-f001:**
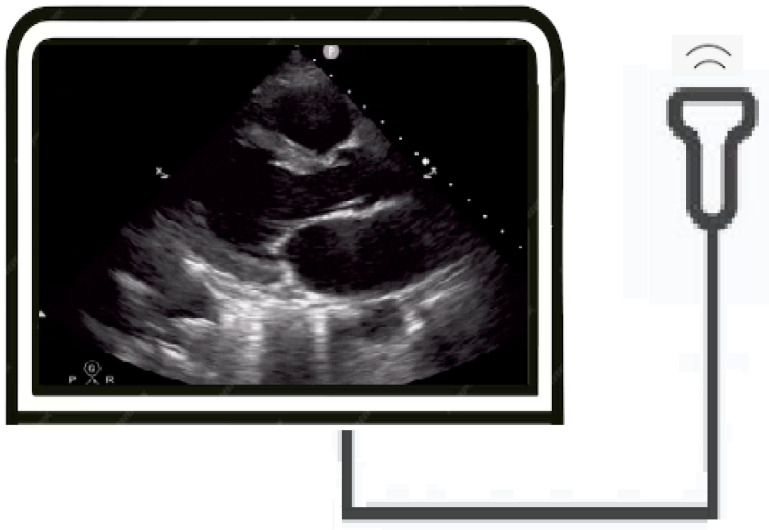
Echocardiography technique.

**Figure 2 medicina-60-01884-f002:**
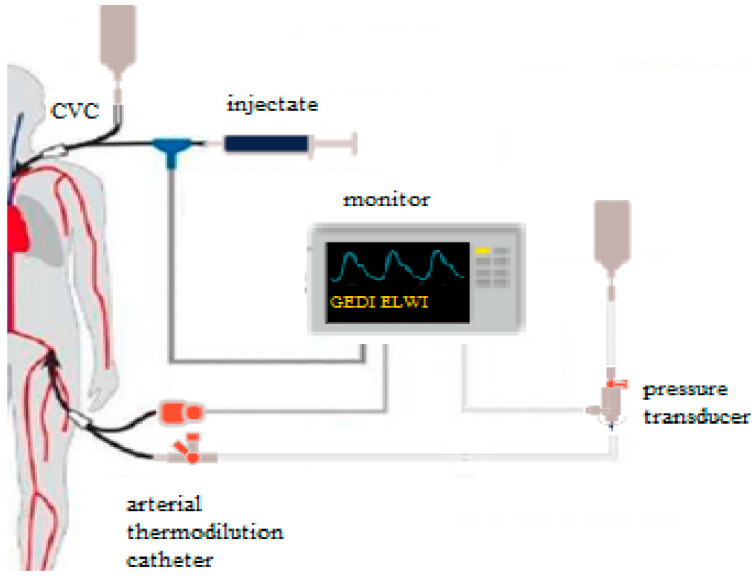
PiCCO technique.

**Table 1 medicina-60-01884-t001:** Cardiogenic shock phases.

	Pre-Shock	Shock	Refractory Shock
SBP	90–100 mmHg	<90 mmHg	<90 mmHg
Heart rate	70–100 bpm	100–120 bpm	>120 bpm
Mental status	Normal	Altered	Absent
Serum lactate	<2 mEq/L	2–4 mEq/L	>4 mEq/L
Cardiac index	2–2.2 L/min/m^2^	1.5–2 L/min/m^2^	<1.5 L/min/m^2^
LVEDP	<20 mmHg	20–30 mmHg	>30 mmHg
Vasopressor therapy	0/1 low dose	1 moderate to high dose	≥2

**Table 2 medicina-60-01884-t002:** Comparison between cardiac ultrasound and PiCCO.

Technique	Echocardiography	PiCCO
Advantages	Non-invasive (transthoracic). Can be repeated with no additional risk. Portable and rapid to implement. Information about the etiology of shock. Distinguishes right from left ventricular failure at the origin of low CO. Provides additional hemodynamic information: left ventricle diastolic properties and filling pressures, left ventricular obstruction, left-sided valvulopathy, tamponade. Helps identification of endocarditis and purulent pericarditis.	Calibrated continuous monitoring of CO (in addition to intermittent CO measurement by trans-pulmonary thermodilution). Detects rapid changes in stroke volume. Useful measurement of parameters: EVLW, GEDV, GEF, CFI. Not operator dependent.
Disadvantages	Not continuous monitoring. Operator dependent. Requires training period. Depends on adequate image quality.	Invasive; requires central venous and intra-arterial access. Catheter insertion-related complications. Hematoma. Infections. Requires external calibrations. Fails to differentiate between left and right ventricular failure. Fails to identify causes of low CO (left ventricular obstruction, severe valvulopathy, tamponade). Pulse contour analysis unreliable in arrhythmias, aortic valve pathology.

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
