# Peer review of "PiCCO or Cardiac Ultrasound? Which Is Better for Hemodynamic Monitoring in ICU?"

_medicina, 2024, doi:10.3390/medicina60111884_

Round 1

Reviewer 1 Report

Comments and Suggestions for Authors

Dear authors,

The manuscript contains 12 pages. The content of the manuscript is consistent with the aims and the scope of the “Medicina” journal.

The manuscript contains: Title, Author list, Abstract, Keywords, Introduction, Conclusions and References.

The title of the manuscript is clearly and precisely formulated and unambiguously refers to the content of the manuscript.

Abstract contains 119 words. There are 6 keywords; only one key word is according to MeSH terminology.

The introduction. In the introduction, the authors of the manuscript state the importance of advanced hemodynamic monitoring of critically ill patients. Also, they give definitions of septic and cardiogenic shock. I would kindly ask the authors to correct the citation of the literature in the text of the manuscript (lines 70, 77).

Further on, the text of the manuscript is divided into the following subheadings: 1. Septic shock and cardiogenic shock, 2. Septic shock and cardiogenic shock, 3. Cardiac ultrasound, 4. Pulse index Continuous Cardiac Output measurements (PiCCO), 5. Conclusion, 6. References.

In the subheading Septic shock and cardiogenic shock some parts of the text are repetitive from “Introduction” (paragraph 4 of the introduction and paragraph 1in septic shock…). Paragraph 2: pathophysiology – is that authors personal interpretation of the pathophysiology OR it is taken from literature, since there are no references at the end of the paragraph? I would kindly ask the authors to correct the citation of the literature in the text of the manuscript (line 114). Paragraph 5 (lines 124-127): is that authors personal interpretation OR it is taken from literature, since there are no references at the end of the paragraph (line 127)? Some parts of the text in paragraph 4 are repetitive from “Introduction” – please consider revision of the text. I would kindly ask the authors to correct the citation of the literature in the text of the manuscript (line 189). Did you create table 1 or you adapt it or take it from another bibliographic source? I would kindly ask the authors to correct the citation of the literature in the text of the manuscript (lines 204, 209, 216, 229, 230).

Subheading Cardiac ultrasound. Paragraph 1: (lines 243-247): is that authors personal interpretation OR it is taken from literature, since there are no references at the end of the paragraph (line 247)? ? I would kindly ask the authors to correct the citation of the literature in the text of the manuscript (lines 268, 274, 288, 291, 303, 312, 318). In line 305 you recalled on Guidelines (according to guidelines …); I can’t find the citated guidelines – on which guidelines are you referring?

Subheading Pulse index Continuous Cardiac Output measurements (PiCCO). I would kindly ask the authors to correct the citation of the literature in the text of the manuscript (lines 338, 341, 347, 356, 367, 373, 376, 387, 398, 408, 415, 422, 428, 436, 442, 449).

Conclusions: conclusions are derived from the text.

References: There are 25 references; 15 references are from 2020 onwards. PLEASE correct the References according to the Reference List and Citations Style Guide for MDPI Journals. https://mdpi-res.com/data/mdpi_references_guide_v5.pdf

Comments on the Quality of English Language

It is necessary to do proofreading of the text in English.

Reviewer 2 Report

Comments and Suggestions for Authors

This is an interesting narrative review attempting to describe the strengths and limitations of cardiac ultrasound vs PiCCO for management of hypotension/shock.  Although this review has much potential, it was very difficult to follow for all of the following reasons: (1) many grammatical errors (see many examples below in the “Comments on Quality of English” section); (2) excessive wordiness that obfuscates the meaning of many sentences (e.g., Lines 149-151); (3) non-linear presentation of ideas; and (4) redundancy (many sections of the paper duplicate text from other parts of the paper).  This paper may eventually be publishable, but first it needs significant revision just to make it readable.  I would recommend review by a native English speaker as a starting point.

Lines: 149-151: I do not understand what the authors are trying to say in either of these 2 sentences.  Please reword this text to clarify what you are trying to say.

Lines 244-245: please add a reference for whatever consensus guideline you are referring to.

Lines 261: the term “2D M-mode echocardiography” is non-standard.  Are you referring to greyscale ultrasound in general (i.e., a combination of B-mode and M-mode)?

Line 264: “TDI” stands for tissue Doppler imaging, not “advanced Doppler techniques.”  Stated another way, TDI is an example of advanced Doppler techniques.

Lines 269-270: This is a controversial statement provided with no reference.  I do not think there is evidence that, “Similar, if not better CO measurements can be obtained with echocardiography rather than using an invasive technique.”  In my experience, quite the opposite is true: cardiac ultrasound is highly capable of generating misleading data about stroke volume and cardiac output because ultrasound is so operator- and angle-dependent.

Lines 448-449: what are the risks associated with “cannulation and ongoing use”?

Comments on the Quality of English Language

Lines 32-35: This is a run-on sentence (2 independent clauses).

Lines 48-51: This is a run-on sentence (2 independent clauses).

Lines 52-55: Awkward wording.   Reads like a run-on-sentence type of error.

Lines 79-82: Awkward wording.   Reads like a run-on-sentence type of error.

Lines 99-100: This is a run-on sentence (2 independent clauses).

Lines 100-103: Awkward wording.   Reads like a run-on-sentence type of error.

Lines 250-253: This is a run-on sentence (2 independent clauses).

Lines 282-288: This is a run-on sentence (2 independent clauses).

Round 2

Reviewer 1 Report

Comments and Suggestions for Authors

Dear Authors,

PLEASE correct the References and citating the references according to the Reference List and Citations Style Guide for MDPI Journals. https://mdpi-res.com/data/mdpi_references_guide_v5.pdf

Comments on the Quality of English Language

It is necessary to do proofreading of the text in English.

Author Response

Please find highlighted changes in the pdf attached.
